# Contribution of Tumor-Derived Extracellular Vesicles to Malignant Transformation of Normal Cells

**DOI:** 10.3390/bioengineering9060245

**Published:** 2022-06-04

**Authors:** Daria S. Chulpanova, Tamara V. Pukhalskaia, Albert A. Rizvanov, Valeriya V. Solovyeva

**Affiliations:** Institute of Fundamental Medicine and Biology, Kazan Federal University, 420008 Kazan, Russia; daschulpanova@kpfu.ru (D.S.C.); tvpukhalskaya@kpfu.ru (T.V.P.); albert.rizvanov@kpfu.ru (A.A.R.)

**Keywords:** extracellular vesicles, carcinogenesis, tumor microenvironment, tumor heterogeneity

## Abstract

Tumor-cell-derived extracellular vesicles (EVs) are known to carry biologically active molecules of parental cells, which can actively modulate the tumor microenvironment. EVs produced by tumor cells play significant roles in the development and maintenance of tumor growth, metastasis, immune escape, and other important processes. However, the ability of EVs to induce the transformation of normal cells has hardly been investigated. This review discusses studies that describe the ability of tumor-cell-derived EVs to alter the metabolism and morphology of normal cells, causing changes associated with malignant transformation. Additionally, the horizontal transfer of oncogenes through EVs of tumor cells and the induction of epigenetic changes in normal cells, which leads to genomic instability and subsequent oncogenic transformation of normal cells, are also discussed.

## 1. Introduction

It is known that tumor transformation of cells requires a gradual accumulation of multiple genetic changes [1]. Such changes can be caused by various external carcinogenic factors, including smoking, alcohol, sunlight, ionizing radiation, viruses, etc. [2], as well as internal factors such as errors in replication and damaged DNA repair processes [3]. It is likely that a combination of extrinsic and intrinsic factors leads to genetic instability and tumor heterogeneity, which are major barriers to cancer treatment. The heterogeneity of mutations in different tumor cells can make a tumor resistant to many chemotherapeutic agents. In addition, the genetic instability of many tumor cells allows them to generate new mutant clones that are resistant to chemotherapy, immunotherapy, and even radiation [4].

However, the appearance of newly transformed cells in the tumor population can also be induced by cancer cells themselves. A number of studies have shown that the cultivation of normal cells with cancer cells leads to the appearance of malignant characteristics in healthy cells, including enhanced proliferation and migration, shift to a mesenchymal phenotype, as well as the development of resistance to γ-irradiation [5]. Furthermore, cell-free chromatin released from dying cancer cells can rapidly enter neighboring normal cells, integrate into the genome, and cause DNA damage and inflammation both in vitro and in vivo [6]. It is important that all these changes in normal cells could be induced without any direct cell–cell contact [5,7]. However, the mechanism of transfer of signaling molecules and DNA that trigger oncogenic transformation is not completely clear. Extracellular vesicles (EVs) released by cancer cells can be one of the key players in this process. EVs are spherical membrane structures capable of transporting molecules of parent cells both on the surface and inside the EVs [8]. EVs can be divided into three groups depending on their biogenesis: exosomes, microvesicles (MVs), and apoptotic bodies (ABs). Exosomes are small spherical structures 40–100 nm in diameter, which are formed as part of the endocytosis pathway. Exosomes are stable in biological fluids and are small enough to pass through the blood–brain barrier. MVs are from 100 to 1000 nm in diameter and are released by budding directly from the cell membrane. Unlike exosomes and MVs, which are continuously produced by cells, ABs are produced as part of the fragmentation process during apoptosis [9]. It has already been shown that EVs secreted by tumor cells play a significant role in the development and maintenance of tumor growth. Exosomes are able to fuse with epithelial cells in the tumor microenvironment (TME) and release their content into intracellular space to eventually stimulate cell proliferation and the formation of new vessels, which supports tumor growth [10]. Moreover, EVs derived from tumor cells are able to suppress the cytotoxic activity of immune cells in TME through PD-L1 and stimulate fibroblast differentiation into the tumor-associated phenotype, thereby contributing to the tumor escape from the immune response [11]. EVs can also transfer functional PD-L1 to other cells that have little or no expression of PD-L1, thereby enhancing the overall immunosuppressing potential of the tumor [12]. In addition, tumor-cell-derived EVs are able to migrate to distant foci in the body and cause morphological and biochemical changes, thereby preparing niches for metastases [13].

However, the ability of EVs to induce the transformation of normal cells has been little studied and is likely another mechanism for the formation of tumor heterogeneity. In this review, we discuss the currently available data on the ability of tumor-cell-derived EVs to induce carcinogenesis in normal cells, as well as the molecular mechanisms that cause metabolic and structural changes in normal cells.

## 2. Influence of EVs of Tumor Cells on the Normal Cells

The mechanisms of modulation of the genotype and functions of normal cells directly depend on the content of the EVs. It has already been shown that proteins entering the recipient cell remain functionally active [14]. Transferred through EVs mRNA can be translated into the functional protein, changing the proteomic profile of the target cell. Moreover, transferred miRNAs can also play an important role in the proteomic signature, as they modulate biological events at the transcriptional and post-transcriptional levels. It has been shown that even small changes caused by the transferred RNAs and proteins affect the metabolism of normal cells [15]. The effect of the biologically active molecules transferred through tumor-cell-derived EV molecules into normal cells is well-supported by the process of pre-metastatic niche formation. For example, tumor-cell-isolated EVs can upregulate the expression of vascular endothelial growth factor (VEGF) and matrix metalloproteinase 2 (MMP2) in endothelial cells, thereby stimulating angiogenesis and vascular permeability in the pre-metastatic niche [16,17]. Moreover, tumor EVs can convert normal fibroblasts into tumor-associated fibroblasts, which also leads to metastatic niche formation and promotes cancer metastasis [18,19]. However, EVs isolated from tumor cells can also alter the metabolism of normal cells of the same tissue type. The existing studies show that tumor-cell-derived extracellular vesicles can increase proliferation rate, induce drug resistance, alterations in endoplasmic reticulum homeostasis, and transition toward mesenchymal phenotype in normal cells [20,21]. Moreover, tumor vesicles can induce genomic instability in healthy cells that can lead to the acquisition of new mutations and malignant transformation [22] (Figure 1).

### 2.1. Increased Proliferation Rate

Tumor cell vesicles entering normal cells can also induce a number of changes in cell properties that are associated with malignant transformation. One of these properties is increased proliferation rate, which indicates healthy cell transformation into cancer cells. It has been shown that EVs isolated from primary prostate cancer, as well as from the plasma of patients with prostate cancer, induce an increase in the proliferation of normal prostate cells by 40% and 60%, respectively, compared with EVs isolated from benign prostatic hyperplasia and plasma of healthy people. The authors of the study attribute this to an increase in the expression of the vimentin (*VIM*) and N-cadherin (*N-cad*) genes and suppression of the E-cadherin (*E-cad*) gene after EV treatment, which indicates a shift in the phenotype of normal cells toward mesenchymal [20]. The effect of EVs from a tissue sample from a patient with a colon tumor on anchorage-independent growth of human normal colon fibroblast cell line 1459 was also shown. Co-cultivation of EVs and healthy cells resulted in a significant increase in normal cell proliferative activity, which was likely mediated by vesicle transfer of the 14-3-3 zeta/delta protein [23]. The same effect was found in normal NIH 3T3 human fibroblasts after cultivation with EVs from MDA-MB-231 triple-negative breast cancer cells or U87 glioblastoma cells. However, in this study, the authors found that such an effect on anchorage-independent growth of fibroblasts was mediated by the transfer of the protein cross-linking enzyme tissue transglutaminase (tTG) [24]. Transfer of the KRAS protein by exosomes of DKO-1 tumor cells also resulted in the increase in the normal DKs-8 cell ability to form anchorage-independent colonies in soft agar [15]. EVs can also carry proteins from oncogenic viruses. For example, Epstein–Barr virus (EBV)-infected B cells transferred the viral latent membrane protein 1 (LMP1) to normal B cells, which enhanced cell proliferation and drove B-cell differentiation toward a plasmablast-like phenotype [25].

In addition to the effect of functionally active proteins, the proliferation of healthy cells can also be mediated by various RNAs transferred by vesicles of tumor cells. Thus, the transfer of circular RNAs from transformed cells can also affect the proliferation of normal cells. It has been shown that circRNA_100284, which is transferred by exosomes from arsenite-transformed human hepatic epithelial (L-02) cells to normal L-02 cells, can inhibit the G1/S transition in the cell cycle, thereby stimulating the proliferation of normal cells [26]. The ability of arsenite-transformed cells to alter the metabolism of normal cells has also been confirmed by other studies. Thus, transformed human bronchial epithelial (HBE) cells were able to carry miR-21, which is activated in arsenite-transformed HBE cells, inside exosomes. The transfer of miR-21 into normal HBEs led to the downregulation of *PTEN* expression, which led to a twofold increase in the proliferation rate [27]. Interestingly, not all tumor cells are able to transform normal cells by transferring biologically active molecules through the EVs. It was shown that EVs isolated from MCF-7 and MDA-MB-231 breast cancer cells did not affect the properties of normal MCF10A breast cells, while HCC1806-derived EVs stimulated the proliferation of MCF10A. Analysis of the gene expression profile and miRNA levels showed that HCC1806-EVs altered the expression of a number of genes associated with the phosphatidylinositol 3-kinase (PI3K)/protein kinase B (AKT) and mitogen-activated protein kinase (MAPK) cell proliferation pathways in MCF10A cells. Changes in the expression of 70 miRNAs associated with pathways activated in cancer cells were also found [21]. In addition, the effect of EV-transported miRNAs on the proliferation of normal cells is also confirmed in vivo. For example, MDA-MB-231-derived EVs containing several miRNAs, as well as the Dicer protein required for their processing, significantly increased the proliferation and viability of MCF10A cells, which allowed these cells to form tumors in nude mice [28]. The described data clearly indicate the transfer of biologically active molecules through EVs triggers a large number of processes in the recipient cells, which can affect cell proliferation, as well as other properties.

### 2.2. Induction of Chemoresistance

It is known that tumor cells resistant to chemotherapeutic drugs mediate the formation of chemoresistance in sensitive cells due to the transfer of miRNAs and other molecules through exosomes [4]. For example, there is a study using prostate carcinoma cell lines DU145 and RC1, the former of which was sensitive to camptothecin, while the latter was not. Co-cultivation of sensitive DU145 cells with EVs derived from resistant RC1 cells has been found to produce the same level of resistance in the DU145 cell lines [29]. Apparently, EVs of tumor cells can also modulate the resistance of normal cells. The addition of EVs isolated from triple-negative HCC1806 breast cancer cells to normal MCF10A cells has induced resistance to adriamycin and docetaxel. It is likely that chemoresistance is a result of altered expression of genes associated with apoptosis activation, such as epidermal growth factor (*EGF*), *RRAS*, and *MAPK3* [21]. Although the modulation of resistance to chemotherapeutic drugs has been mostly shown for the already transformed cells that do not yet have resistance, a small number of studies confirm that similar effects of the drug resistance development can also be observed in non-malignant cells after their interaction with tumor-derived EVs.

### 2.3. Endoplasmic Reticulum Homeostasis Alteration

Released by tumor cells EVs were also able to change the homeostasis of the endoplasmic reticulum (ER) of normal cells. Incubation of non-malignant human SV-HUC urothelial cells with tumor-cell-derived EVs led to the induction of ER stress and triggered an unfolded protein response (UPR). In normal cells, in case of prolonged ER stress activation of UPR may initiate apoptotic cell death via the upregulation of the C/EBP homologous protein (CHOP) [30]. Indeed, CHOP expression was activated in the normal cells treated with tumor EVs for 8 weeks. However, prolonged treatment of SV-HUC cells with tumor vesicles (13 weeks) led to a shift in metabolism toward an inositol-requiring enzyme 1 (IRE1)-mediated survival pathway instead of CHOP-mediated apoptosis induction [31]. At the same time, the level of CHOP was significantly upregulated in normal cells after long-term cultivation with tumor EVs, which is also observed in superficial and invasive bladder cancer relative to normal bladder tissue [32]. Changes in ER homeostasis also led to activation of UPR-induced inflammatory response, acquired resistance to growth inhibition by cell-to-cell contact, and increased anchorage-independent growth [31]. Thus, cancer EV exposure might reprogram normal cells to switch to the survival IRE1-mediated pathway and inhibit CHOP-mediated apoptosis, thereby promoting their survival. At the same time, chronic UPR-associated inflammation can contribute to normal cell transformation through multiple mechanisms, such as genome instability, which is discussed below.

### 2.4. Induction of EMT Transition

It is also worth noting that many studies noted that tumor-cell-derived EVs trigger phenotypic and functional changes in normal cells that lead to a transition from epithelial to mesenchymal phenotype (EMT), the process which allows the solid tumors to become more malignant, increasing their invasiveness and metastatic activity [33]. For example, incubation of keratinocytes with EVs derived from metastatic melanoma cells activates the AKT/mammalian target of rapamycin (mTOR) and extracellular-signal-regulated kinase (ERK) signaling pathways, thereby increasing cell migration ability. In addition, tumor-cell-derived EV-treated keratinocytes had a stem-like phenotype and increased expression of *SNAI1*, *KLF*, and CD133 factors that trigger and maintain EMT [34]. Migration-enhancing morphological changes—namely, an increase in cell length and the number of filopodia—were also observed in normal lung epithelium BEAS-2B cells after the addition of exosomes from non-small lung carcinoma cell line H1299 [35]. Such changes in the migration ability of cells and their transition to the mesenchymal phenotype are closely associated with changes in the expression of cadherins. Transfer of annexin A1 protein by exosomes from thyroid squamous cell carcinoma SW579 cells induces EMT in human Nthy-ori3-1 thyroid follicular epithelial cells due to the decrease in the p-Smad2 and N-cadherin protein expression level and an increase in the E-cadherin protein expression level [36].

### 2.5. Horizontal Transition of Oncogenes

Tumor formation involves the accumulation of a number of mutations in a cell genome essential for malignant transformation. Often, mutations occur in certain genes associated with carcinogenesis–oncogenes. The phenomenon of horizontal transfer of oncogenes by EVs and other ways has already been shown for tumor cells. For example, glioma cells expressing an oncogenic form of the EGF receptor known as EGFRvIII can transfer it to other tumor cells with normal EGFR via vesicular transport [37]. KRAS mutant form transfer from the blood plasma of patients with colon cancer to normal mouse fibroblasts, which can also be mediated by vesicular transport, is also described [38]. Fragments of genomic DNA that have entered the recipient cell are able to be transcribed, which leads to the synthesis of the corresponding protein. In a study by Cai et al., it has been shown that EV-mediated transport of the breakpoint cluster region (BCR)/Abelson murine leukemia viral oncogene (ABL) hybrid gene from K562 donor cells to HEK293 recipient cells results in the expression of mRNA and protein of the BCR/ABL hybrid gene. In addition, the hybrid gene was also transferred from K562 cells to neutrophils from human peripheral blood, causing a decrease in their phagocytic activity [39]. The oncogenic potential of BCR/ABL-carrying EVs has also been confirmed in vivo in immunodeficient NOD/SCID mice, where administration of BCR/ABL EVs resulted in the formation of chronic myelogenous leukemia (CML)-like phenotype, with characteristics of CML such as feeble, febrile, thin, and with splenomegaly and neutrophilia but with reduced neutrophil phagocytic activity [40].

Normal cells have also been shown to be able to uptake ABs carrying the oncogenes of donor cells. For example, the internalization of fragmented DNA containing human H-RASV^12^ and C-MYC genes induced loss of contact inhibition in mouse embryonic fibroblasts. Moreover, injection of SCID mice with the fibroblasts transformed after interaction with ABs led to the formation of tumors, which indicates the development of the full tumorigenic potential [41]. The ability to carry H-RASV^12^ has also been shown for smaller EVs. Thus, for example, mouse fibroblasts acquired increased viability after H-RASV^12^ transfer from immortalized epithelial cells transfected with a V12 mutant. However, treatment of normal epithelial cells with EVs from mutant cells did not increase the incidence of tumor formation in SCID mice [42]. This also includes the transfer of viral DNA of viruses from apoptotic tumor cells to normal ones, which results in carcinogenesis induction in normal cells. Human primary fibroblasts have actively uptaken HeLa-cell-derived ABs containing integrated high-risk human papillomavirus (HPV) DNA that causes cervical cancer [43]. At the same time, the expression of viral genes in fibroblasts led to the dysregulation of the p53/p21 pathway, increased proliferation rate, polyploidy, and anchorage independence growth [44]. Using the same mechanism of apoptotic body uptake, genes of EBV *eber* and *ebna1* can be transferred and efficiently expressed in normal human fibroblasts [45].

### 2.6. Induction of Genomic Instability

Another important discovery is that EVs, besides modulating the metabolism of normal recipient cells, also induce genomic instability, the most important hallmark of cancer, that leads to an increase in genetic alterations [46]. It has been shown that incubation of leukemia cells with K562-derived MVs leads to an increased DNA breakage in normal cells, which may be mediated by increased levels of reactive oxygen species that cause oxidative DNA damage [22]. In addition, tumor cell EVs are able to induce micronucleus formation in normal endothelial cells, as well as increase the level of DNA damage response markers such as phosphorylation of histone γH2AX [47]. In addition to genomic effects, MVs also induce epigenetic changes in recipient cells, which also leads to genomic instability. In cells treated with K562-MVs, hypermethylation of promoters of tumor suppressor genes *P53* and *RIZ1* was observed, which are often regulated by methylation [22].

## 3. Conclusions

To date, the fact that tumor-cell-derived EVs can significantly modulate tumor development is beyond doubt. Numerous studies confirm the ability of EVs to modulate angiogenesis, invasion, and tumor localization by changing the behavior of surrounding non-malignant cells. At the same time, there are not so many studies dedicated to the possibility of EVs inducing carcinogenesis in normal cells. However, the wide variety of biological molecules that contribute to the transformation of normal cells suggests that the contribution of such a factor as vesicular transport in tumorigenesis cannot be ignored. Current studies show that EVs released by tumor cells can significantly change the metabolism of normal cells, enhancing their proliferation, and anchorage-independent growth, and also stimulate the development of chemoresistance. The significance of these changes is confirmed by the fact that normal cells with increased carcinogenic potential become capable of forming tumors in immunodeficient mice. However, it is known that, for malignant transformation, a normal cell must accumulate a number of oncogenic mutations. EV-transferred molecules are capable of altering normal cell metabolism but are unable to induce stable mutations, lack the potential to self-replicate, and their effects on cells are inherently self-limiting due to their inevitable degradation and dilution. Indeed, several of the studies described above have shown that when cancer EV exposure is terminated, normal cell metabolism can be restored after some time.

However, the existence of the phenomenon of field cancerization [48], in which normal cells around the tumor tissue undergo carcinogenic changes, indicates that the influence of tumor cells on the surrounding normal tissue is not limited to the regulation of metabolism. A number of studies indicate that the transfer of mutant DNA through ABs and MVs may be one of the possible mechanisms of genetic instability and genetic diversity in tumors. It is possible that the horizontal transfer of tumor cell DNA and their subsequent transcription and translation lead to the accumulation of genetic changes essential for the malignant transformation of normal cells. However, in our opinion, the most probable mechanism is the induction of genetic instability due to altered metabolism and DNA oxidative damage, as well as the regulation of the expression of oncosuppressor genes due to methylation. This area requires more careful attention from researchers in order to come to unambiguous conclusions about the mechanisms of carcinogenesis induction in normal cells surrounding the tumor and the degree of contribution of tumor-cell-derived EVs to this process.

Since the effect of field cancerization has been described for a large number of tumors, such as bladder cancer and prostate cancer, the modulation of EV secretion in these tumors may be an important part of therapy that will allow preventing further malignancy of the nearby field and reduce the chance of disease recurrence. At the same time, whereas EVs carry out hundreds of different biologically active molecules into normal cells, it is possible that modulation of EV secretion could be a more advantageous approach to control supporting TME as well as preventing the spread of tumor drug resistance. Suppression of EV released by tumor cells can be achieved clinically with drugs such as amiloride, which blocks both biogenesis and macropinocytosis of EVs, as well as imipramine, calpeptin, manumycin A, etc. [49,50]. Moreover, the antitumor effect of imipramine is currently under investigation in clinical trials in patients with ER-positive and triple-negative breast cancer (NCT03122444, http://clinicaltrials.gov, accessed on 17 April 2017) and recurrent glioblastoma (NCT04863950). However, the results have not yet been reported.

Thus, the contribution of EVs to the malignant transformation of normal cells requires the attention of researchers, and in order to successfully investigate this area, future studies should be aimed at suppressing EV biogenesis and subsequent analysis of changes in the phenotype of normal cells.

## Figures and Tables

**Figure 1 bioengineering-09-00245-f001:**
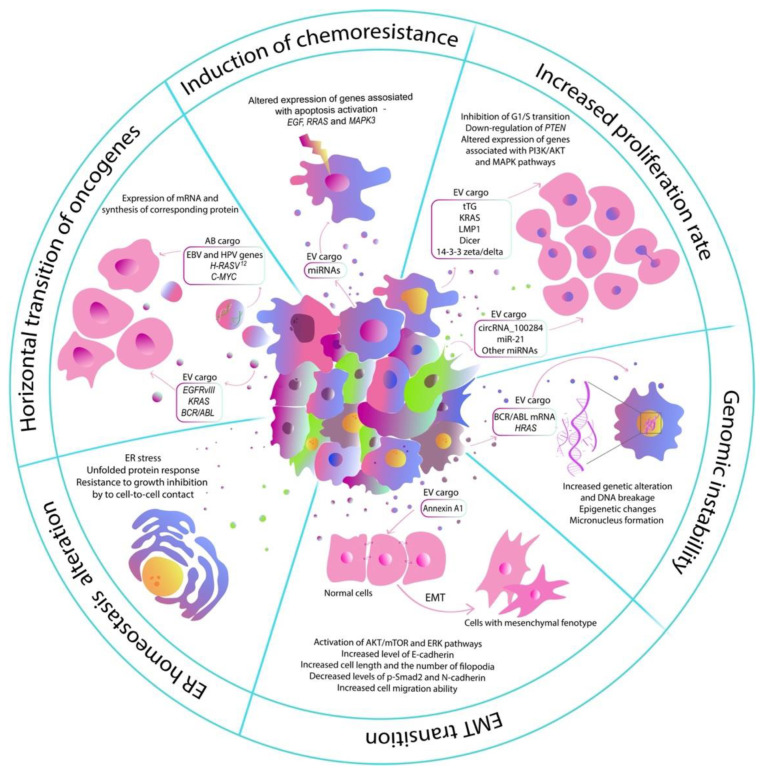
The influence of tumor-cell-derived extracellular vesicle on the metabolism, structure, and epigenetic profile of normal cells.

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
