# Peer review of "Contribution of Tumor-Derived Extracellular Vesicles to Malignant Transformation of Normal Cells"

_bioengineering, 2022, doi:10.3390/bioengineering9060245_

Round 1

Reviewer 1 Report

The authors report contribution of tumor-derived extracellular vesicles to malignant transformation of normal cells.

  1. The authors should describe the relationship between extracellular vesicles and PD-1 or PD-L1 in a text.
  2. The authors should provide the information of clinical trial list and describe in a text. It will be benefit for the reader.

Author Response

Thank you for your comments. We have corrected your comments point by point within the manuscript accordingly (your comments are in bold text and our responses are in ordinary type):

  1. The authors should describe the relationship between extracellular vesicles and PD-1 or PD-L1 in a text.

We have added information on the interaction of EVs from cancer cells carrying PD-L1 with the immune cells (Page 2, Lines 53-58).

  1. The authors should provide the information of clinical trial list and describe in a text. It will be benefit for the reader.

We have pointed existing clinical trials dedicated to the analysis of the antitumor activity of drugs that prevent vesicle formation (Page 7, Lines 291-294).

Reviewer 2 Report

The article is well designed and clarified  about tumor cell-derived extracellular vesicles (EVs), and its role of  tumor microenvironment.

Author Response

Thank you for your comment. We have also added the information on the effect of tumor cells on the normal cell metabolism (Page 1, Lines 31-40) and how EVs induce unfolded protein response in ER of normal cells (Page 4, Line 161 – Page 5, Line 178).

Reviewer 3 Report

This paper comprehensively reviewed how the EV affect normal cells in vitro. The molecular mechanisms are also explained. The review provided good further directions which are helpful for researchers. The comments for revision are given below.

  1. Please include discussion on different effects of culturing cells with isolated EV and the co-culture with cancer cells in section 2.1.

  1. Please explain more on how the cited study in section 2.3 showed the “change of homeostasis of the endoplasmic reticulum”. Please give more details and connect them logically to make the listed observations support this conclusion.

  1. Please add a paragraph in section 2 to summarize possible ways that EV affects normal cells and leads to tumor metastasis.

Author Response

Thank you for your valuable comments. We have corrected your comments point by point within the manuscript accordingly (your comments are in bold text and our responses are in ordinary type):

  1. Please include discussion on different effects of culturing cells with isolated EV and the co-culture with cancer cells in section 2.1.

We have added information on how co-cultivation with cancer cell affects properties of normal cells and if it induces oncogenic transformation in normal cells (Page 1, Lines 31-40).

  1. Please explain more on how the cited study in section 2.3 showed the “change of homeostasis of the endoplasmic reticulum”. Please give more details and connect them logically to make the listed observations support this conclusion.

We described in details how EVs from cancer cells induce stress and unfolded protein response (UPR) in endoplasmic reticulum of normal cells, which leads to survival instead of death of affected cells. As a result, chronic UPR-associated inflammation can contribute in normal cell transformation through multiple mechanisms, such as genome instability (Page 4, Line 161 – Page 5, Line 178).

  1. Please add a paragraph in section 2 to summarize possible ways that EV affects normal cells and leads to tumor metastasis.

We have added information on the effect of EVs on the pre-metastatic niche formation and metastasis in Page 2, Lines 76-87.

Round 2

Reviewer 1 Report

none